# Mesenchymal Stem Cells and Exosomes: A Novel Therapeutic Approach for Corneal Diseases

**DOI:** 10.3390/ijms241310917

**Published:** 2023-06-30

**Authors:** Basanta Bhujel, Se-Heon Oh, Chang-Min Kim, Ye-Ji Yoon, Young-Jae Kim, Ho-Seok Chung, Eun-Ah Ye, Hun Lee, Jae-Yong Kim

**Affiliations:** Department of Ophthalmology, University of Ulsan College of Medicine, Asan Medical Center, 88, Olympic-Ro, Songpa-Gu, Seoul 05505, Republic of Korea; basantabhujel86@gmail.com (B.B.); 72004218osh@gmail.com (S.-H.O.); kcm8821@naver.com (C.-M.K.); yejii0849@gmail.com (Y.-J.Y.); dudwo991023@naver.com (Y.-J.K.); chunghoseok@gmail.com (H.-S.C.); yeyeyeyeye@gmail.com (E.-A.Y.); yhun777@hanmail.net (H.L.)

**Keywords:** cornea, mesenchymal stem cells, exosomes, corneal diseases, corneal regeneration

## Abstract

The cornea, with its delicate structure, is vulnerable to damage from physical, chemical, and genetic factors. Corneal transplantation, including penetrating and lamellar keratoplasties, can restore the functions of the cornea in cases of severe damage. However, the process of corneal transplantation presents considerable obstacles, including a shortage of available donors, the risk of severe graft rejection, and potentially life-threatening complications. Over the past few decades, mesenchymal stem cell (MSC) therapy has become a novel alternative approach to corneal regeneration. Numerous studies have demonstrated the potential of MSCs to differentiate into different corneal cell types, such as keratocytes, epithelial cells, and endothelial cells. MSCs are considered a suitable candidate for corneal regeneration because of their promising therapeutic perspective and beneficial properties. MSCs compromise unique immunomodulation, anti-angiogenesis, and anti-inflammatory properties and secrete various growth factors, thus promoting corneal reconstruction. These effects in corneal engineering are mediated by MSCs differentiating into different lineages and paracrine action via exosomes. Early studies have proven the roles of MSC-derived exosomes in corneal regeneration by reducing inflammation, inhibiting neovascularization, and angiogenesis, and by promoting cell proliferation. This review highlights the contribution of MSCs and MSC-derived exosomes, their current usage status to overcome corneal disease, and their potential to restore different corneal layers as novel therapeutic agents. It also discusses feasible future possibilities, applications, challenges, and opportunities for future research in this field.

## 1. Introduction

Loss of vision is an expanding global burden that affects not only the individual but also society as a whole. The human eye is a highly complex organ that has a crucial role in our life, but unfortunately, it is susceptible to a broad spectrum of disorders. Despite the challenges, there have been numerous efforts to address a wide range of eye diseases [1]. The current therapeutic approaches for treating corneal disorders include anti-inflammatory drugs, limbal stem cell (LSC) transplantation, and corneal transplantation [2]. However, these modalities are not free from limitations. LSC transplantation has a high risk of immune rejection [3,4]. Anti-inflammatory drugs are not fully capable of suppressing angiogenesis, conjuctivalization, and corneal scarring [5]. Although corneal transplantation is still considered an effective means of restoring vision, graft rejection due to immune responses remains a significant cause for concern. As estimated in the 2010 World Health Organization’s report on visual impairment, about 285 million people of all age groups experience some level of visual impairment, and 39 million people are completely blind [6].

The cornea is one of the main components of the visual system, and it is the prime layer of the eye. It is crucial for vision, as it focuses light and plays a refractive role [7]. Structurally, the human cornea consists of five layers, which include the epithelium, stroma, endothelium, Bowman’s layer, and Descemet’s membrane. Nonetheless, the most significant layers among these are the epithelium, stroma, and endothelium [8]. The pathogenesis of corneal diseases may be attributed to a range of clinical conditions, including traumatic injury, chemical exposure, infections, deterioration due to aging, limbal stem cell deficiency, and various types of corneal dystrophies. This latter phenomenon may lead to defects in the structural and cellular components of the cornea [9]. The corneal functions can be compromised due to the development of corneal scars, haze, opacities, and edema, which may lead to visual deterioration. However, early detection and timely treatment can prevent most cases of corneal blindness.

Corneal transplantation using a healthy donor cornea to replace the damaged cornea has been successfully carried out for 100 years. It is currently considered one of the standard treatment strategies for corneal blindness. The cornea is the most frequently transplanted solid tissue in humans [10]. Although there have been notable improvements in corneal surgery in recent years, there are still challenges concerning the shortage of donor tissue, the short lifespan of allografts, the prolonged use of immunosuppressive drugs, legal and cultural restrictions, and the requirement for specialized surgical skills. A significant proportion of patients cannot afford corneal transplantation due to the high costs associated with the surgery and post-operative care. This is especially problematic given the increasing number of elderly individuals, resulting in significant financial and logistical obstacles, thus creating a global burden [5].

The transplantation of corneas is the most commonly performed type of transplant on a global scale, with roughly 180,000 corneal transplantation surgeries being conducted each year [11,12]. During the year 2012, over 184,000 corneal transplantation procedures were performed across 116 different nations [11]. As stated by the Eye Bank Association of America, the quantity of donated corneas and eye globes has been steadily increasing over the past few years, with a 5.2% increase reported in 2013 relative to 2012 [11]. An estimated 12.7 million individuals around the world are in need of corneal transplants, but the availability of donor tissues varies greatly across different regions. In 2019, in the UK, 4504 corneal transplants were carried out, while in the USA, the number of transplanted corneas was significantly higher, at 85,601 [13]. Nevertheless, the supply of transplantable donor tissue is consistently insufficient, as the demand for it surpasses the availability [11].

In recent decades, cell therapy and tissue engineering approaches have gained in popularity as treatments for some corneal diseases [14]. When it comes to tissue engineering, the cornea is well-suited for regenerative cell therapy using natural or systemic scaffolds, with or without cells, because of its immune-privileged and avascular characteristics [15]. Compared to other organs or tissues, there is a lower probability of rejection of the transplanted cells in the cornea. Today, ophthalmologists and visual scientists are increasingly interested in mesenchymal stem cells (MSCs) due to their competence to regenerate and differentiate, making them a potential alternative treatment option for corneal diseases. MSCs have been suggested to exhibit a therapeutic effect through their paracrine effect, which is mediated by exosomes (Figure 1) [16,17]. Research investigating the mechanism of MSC-based therapies has yielded compelling evidence indicating that exosomes facilitate intercellular communication by transporting various biomolecules, including nucleic acids and proteins, to the recipient cells. Additionally, they are also involved in essential processes such as cell migration and differentiation, and they participate in many physiological and pathological functions [18,19]. The effectiveness and safety of exosomes for curing corneal diseases have garnered attention due to the exosomes’ cell-free nature, which enables them to retain the therapeutic advantages of their parent cells while eliminating the risks associated with stem-based therapies. As a result, the role of exosomes in regenerative medicine is expanding. To conduct this review, we performed a thorough search of PubMed (https://pubmed.ncbi.nlm.nih.gov/ (accessed on 1 May 2023) and Google Scholar using the terms “corneal diseases,” “exosomes,” and “mesenchymal stem cells” as keywords from July 2002 to April 2023.

## 2. Mesenchymal Stem Cells (MSCs)

Stem cells can be categorized into two types depending on their source: embryonic and adult stem cells. Adult stem cells are present in a multitude of organs and tissues within the body, such as skeletal muscles, brain, bone marrow, dental pulp, liver, spinal cord, cornea, adipose tissue, and more. Among these, MSCs, regarded as multipotent progenitor cells, can arise from either embryonic or adult sources [20]. Wharton’s jelly, umbilical cord blood, placenta, and embryo can provide embryonic stem cells, whereas dental pulp, bone marrow, adipose tissue, and other tissues are considered adult sources [21]. Embryonic MSCs contain a large number of primitive phenotypes, more active telomeres, and higher propagation ability compared to stem cells derived from adult tissues [22]. However, to obtain a sufficient amount of embryonic MSCs for therapeutic applications, ex vivo expansion is necessary, which may lead to a decline in their functional activity [23].

Protocols for the in vitro derivation of MSCs from human pluripotent stem cells, such as induced pluripotent stem cells (iPSCs) and embryonic stem cells, have been formulated and implemented [24]. Both in vitro and in vivo, MSCs can differentiate into various mesenchymal lineages, such as osteoblasts, adipocytes, and chondrocytes. Furthermore, they can migrate to injured sites, where they can differentiate and proliferate, secrete various anti-inflammatory and growth factors, promote wound healing, and thus reconstruct the damaged tissues [25].

MSCs play a vital role in the modulation of immune responses via paracrine action and interact with both innate and adaptive immune cells [26]. MSCs derived from different sources have different functions. The phenotypic markers CD13, CD73, CD90, CD105, and STRO-1 are expressed by both bone marrow and adipose MSCs, but they have different expression patterns of CD34, CD49d, CD54, and CD106 [27]. MSCs derived from adult tissues like bone marrow show less proliferation, engraftment ability, and differential potential than MSCs derived from birth-associated tissues (cord blood, umbilical cord, placenta, and amnion) [28]. Different proteomic profiles can be observed in the MSCs derived from the umbilical cord, Wharton’s jelly, or cord blood. MSCs derived from the umbilical cord show ameliorated results in musculoskeletal tissue engineering [29]. Similarly, the MSCs derived from adult adipose tissue sources also demonstrate variations in functionality [30]. MSCs derived from subcutaneous fat tissue have quicker tissue proliferation than those derived from the omental region [31]. MSCs are identified by the existence of specific markers such as CD90, CD73, CD71, CD44, CD105, and CD271, but they do not express hematopoietic markers (CD14, CD34, and CD45) nor do they express stimulant molecules (CD86, CD40, and CD80) [32]. Studies have demonstrated that MSCs produce a diverse array of exosomes, which perform various tasks like immune modulation, repairing damaged tissue, and downregulating inflammation via paracrine action [33,34].

MSCs exhibit potential immunomodulatory effects, which contribute to their therapeutic potential and ability to evade rejection both in vivo and in vitro. MSCs, considered to be immune-privileged cells, diminish the presence of major histocompatibility complex (MHC) class II molecules and co-stimulatory molecules (CD80, CD86, and CD40) on their cell surface [35]. In vitro, MSCs affect the innate immune system by suppressing the maturation and activation of dendritic cells (DCs) as well as the cytotoxicity of natural killer cells. Moreover, they suppress adaptive immune responses by inhibiting the proliferation and secretion of cytokines by T cells, as well as impeding the maturation of B cells [36]. Various soluble factors, such as transforming growth factor-beta (TGF-β), interleukin-6 (IL-6), interleukin-10 (IL-10), matrix metalloproteinases (MMPs), prostaglandin E2 (PGE2), indoleamine-2, 3-dioxygenase (IDO), human leukocyte antigen-G5 (HLA-G) and nitric oxide are involved in the immunosupressive function of MSCs [37]. Moreover, MSCs can suppress the production of interferon-gamma (IFN-γ) by Th1 cells while increasing the production of IL-4 and IL-10 by Th2 cells. This alteration in cytokine expression facilitates the immune response of native CD4+ T cells, promoting a shift toward a Th2-type immune response [38]. In co-culture with native T cells, human MSCs support the differentiation and proliferation of regulatory T cells (Tregs) during mixed-lymphocyte reactions. This effect is mediated through the secretion of prostaglandin E2 (PGE2) and TGF-β. Tregs, which are a specialized subset of T cells, retain their ability to suppress the activation of other T cells and help regulate immune system activity [39]. In addition, microencapsulation within microspheres or hydrogels can also protect MSCs physically from immune recognition and prevent cell aggregation, thereby avoiding any direct contact and reducing the risk of rejection in the transplanted site [40,41].

Even though MSCs have great potential in regenerative medicine, there are concerns about their uncontrolled proliferation or overgrowth, which can lead to unwanted effects. Implementing kill switches in the context of MSCs involves incorporating mechanisms to control and terminate their growth when necessary. One potential approach is to engineer MSCs with genetic circuits that allow for conditional cell death or growth arrest. These circuits can be designed to respond to specific signals or triggers in the cellular environment [42]. Likewise, MSCs can be genetically engineered to control their proliferative capacity and immunomodulatory effects, preventing over-expansion [43]. In addition to this, manipulating cell cycle regulators like p16^Ink4a^ or p21^cip1^ can block cell cycle progression and reduce their proliferation. Additionally, modulating the activity of key regulators, such as retinoblastoma protein (pRB) or p53, can also impact MSCs growth [44]. Hypoxic conditions can slow cell proliferation and promote a quiescent state of MSCs. TGF-β signaling is involved in cell cycle regulation and can affect growth of MSCs. Modulating the activity of TGF-β signaling pathway components, such as SMAD proteins or TGF-β receptors, can help control MSCs proliferation [45]. Incorporating MSCs and growth factors or bioactive molecules within scaffolds can regulate MSC behavior. Furthermore, some scaffold materials, such as collagen and gelatin, have been reported to support controlled MSCs growth and prevent overgrowth [46].

## 3. MSC Mobilization, Migration, and Homing in Corneal Changes

Injury and inflammation are responsible for inducing the mobilization, migration, and colonization of stem cells [25,34]. The mechanism of MSC homing to the sites of injury remains unclear. When the cornea is injured by trauma and infection, specific chemoattractants stimulate endogenous bone marrow MSCs, causing them to mobilize and enter the peripheral blood. To promote wound healing, these circulating MSCs travel to the site of injury in the cornea and attach themselves to it [47]. In a study of a murine alkali-burn model of the injured cornea, the researcher demonstrated the vigorous migration and engraftments into the injured cornea of bone marrow MSCs administered intravenously [48]. Chemokines such as stromal cell-derived factor 1 (SDF-1) and substance P have been identified as regulators of MSC mobilization and recruitment to the cornea [47]. Additionally, MSCs’ ability to locate and bind to target tissues is facilitated by robust mechanisms similar to leukocyte activity, including adhesion mediated by selectin and integrin, transmigration, and passive entrapment [49]. Efforts have been made to upgrade the targeting of MSCs to ocular tissues. The sub-conjunctival injection of MSCs and their co-transplantation with amnion onto the damaged corneal tissue have both resulted in a substantial enhancement [50,51,52,53].

Despite their ability to migrate to the site of injury, it appears that the migration and homing of MSCs are not essential for their therapeutic effects. In one study, the damage caused by inflammation in the cornea was reduced by tumor necrosis factor-stimulated gene/protein-6 (TSG-6) released by MSCs administered systemically, even without MSC engraftment [54]. Furthermore, MSCs have been found to secrete TSG-6, which has been shown to have a positive effect on reducing inflammation and improving cardiac function after a myocardial infarction [55]. A recent study showed that administering human MSCs through the systemic route can diminish inflammatory damage in the cornea by releasing anti-inflammatory substances when prompted by injury signals from the cornea, with no need for engraftment [54]. One potential approach to enhancing the effectiveness of MSCs in treating corneal injuries is to administer them directly to the affected area using methods such as subconjunctival injection, transplantation with an amniotic membrane (AM), or application through a plastic tube. This could result in an increased MSC concentration at the site of the injury, leading to improved outcomes [51,56,57,58].

## 4. Corneal Regeneration with MSCs

There is considerable research indicating that MSCs are capable of reducing inflammation and facilitating the restoration of corneal transparency in the aftermath of ocular injuries [48,57,59]. Despite the fact that MSCs promote angiogenesis in certain tissues, they exhibit an opposing effect in the cornea by inhibiting angiogenesis. This distinctive quality of MSCs represents an intriguing and noteworthy characteristic of these cells [60]. Researchers have explored two primary methods for administering MSC treatment for corneal injuries: intravenous injection and topical application. A number of studies have examined the effectiveness of intravenous injection in repairing corneal injuries [61,62,63]. In a study using adipose-derived stem cells (ADSCs) on an amniotic membrane, the researchers observed a reduction in inflammation, an increase in corneal transparency, and alleviation of the corneal damage [64]. Similarly, the damage to the cornea was partially repaired via the secretion of anti-inflammatory proteins by the injected MSCs [54]. Research has revealed that the culture medium derived from MSCs has the potential to treat corneal ulcers by suppressing inflammation, enhancing cell viability, stimulating proliferation, and modulating the immune response [65]. The therapeutic role of MSCs in different layers of the cornea is shown in Table 1. Figure 2 shows the therapeutic roles of MSCs in regenerating the cornea.

### 4.1. Corneal Epithelial Regeneration with MSCs

The corneal epithelium is fairly uniformly composed of 5–7 layers of stratified non-keratinized squamous cells, each 50 μm thick, which covers the outermost portion of the cornea [77,78]. The maintenance of a healthy corneal epithelial layer is essential for appropriate visual function and translucency of the cornea [79]. The tear film covers the surface of the epithelium, which is made up of a non-keratinized squamous layer that originates from the superficial ectoderm during embryonic development at around 5–6 weeks [78]. This layer has a crucial function in correcting any abnormalities present on the surface of the cornea [6]. Any disruption of the structural soundness of the corneal epithelium due to physical trauma, infection, or LSC deficiency, could lead to damage to the epithelium, corneal inflammation, neovascularization, and opacities. Eventually, this phenomenon can culminate in corneal blindness [80,81]. MSCs can differentiate into cell lineages derived from the neuroectoderm and epithelial cells [82]. An in vitro study co-cultured rabbit bone marrow MSCs and rabbit LSCs and found that the cells displayed a polygonal and cobblestone morphology similar to epithelial cells. Moreover, these cells expressed cytokeratin 3 (CK3), specific to corneal epithelium [82]. Similarly, a study involving co-culture of rat corneal stromal cells (CSCs) and MSCs found that MSCs underwent trans-differentiation into cells that resembled epithelial cells, confirmed through the observation of CK12 expression. Furthermore, in vivo experiments showed reduced corneal opacity and neovascularization grades, indicating the potential of MSCs to improve corneal integrity [53]. When human ADSCs were grown in a culture medium that had been conditioned by corneal epithelial cells, the result illustrated the upregulation of the expression of CK3 and CK12 with an epithelial-like cell appearance [83,84]. An in vivo study on mesenchymal–epithelial transition (MET) was inconclusive. When human BM-MSCs were tissue engineered onto human amnion and then transplanted into rat corneas damaged by alkali, there was a decline in inflammation (as demonstrated by the markers CD45, IL-2, MMP-2) and angiogenesis markers accompanied by reconstruction of the corneal surface [57].

The utilization of small-molecule chemicals has become increasingly popular since they have shown potential in initiating and modifying the cellular changes involved in the transition between mesenchymal and epithelial phenotypes. Furthermore, these small molecules can also regulate cell fate and facilitate the reprogramming of target genes [85,86]. A protocol has been previously reported for generating corneal epithelial progenitors, referred to as MET-Epi, from human ADSCs. This protocol involves an approach that antagonizes both the GSK and transforming growth factor β (TGF β) pathways [47]. These small molecules can penetrate cell membranes, lack immunogenicity, have modifiable dosages, are cost-effective, and expedite standardization. In an investigation utilizing a rat corneal surface alkali injury model, it was observed that the application of ADSCs-derived epithelial progenitors engineered on fibrin gel led to an enhancement of corneal transparency and stability of the surface. The damaged corneal surface was restored with the creation of a multilayered epithelium. Moreover, higher levels of human epithelial cell adhesion molecule (EpCAM) and CK3/12 expressions were noted. The sham group had less reduction of corneal haze and no CK3/12 expression [48].

An alternative surgical approach to corneal epithelial failure, which can arise from severe limbal stem cell deficiency (LSCD), involves the use of cultivated limbal epithelial transplantation (CLET) with either an amniotic membrane or fibrin gel acting as a carrier [87]. A new procedure that involves ex vivo propagation of autologous or allogeneic epithelial stem cells from a limbal biopsy can help stabilize the ocular surface, and it has theoretical advantages over conventional limbal transplantation. A recent study demonstrated the potential of MSC therapy to achieve similar therapeutic results as CLET [88]. Another study compared the safety and efficacy of allogeneic BM-MSC transplantation and allogeneic CLET for the restoration of corneal epithelia in LSCD over a one-year follow-up period, and it was revealed that both treatments were effective and safe [68]. In another study of transplantation of human foreskin-derived mesenchymal stem cells (hMSCs) into alkaline damaged rabbit epithelium, differentiation of these cells into corneal cells and their migration into corneal stroma was observed [89]. Based on the findings of these studies, MSCs may serve as a viable alternative option for treating damaged corneal epithelium and promoting the repair of the ocular surface. Although MSC therapy has shown promise in treating corneal epithelial damage, additional research is necessary to provide evidence for its clinical effectiveness, safety, and long-term stability.

### 4.2. Corneal Stromal Regeneration with MSCs

The corneal stroma, the thickest layer of the cornea, consists of specific extracellular matrix (ECM) elements and collagen fibrils organized into flattened lamellae that run perpendicular to each other. The corneal stromal keratocytes (CSKs) are typically quiescent and are located between collagenous lamellae [90]. The cornea remains transparent, biomechanically strong, and structurally intact due to the precise arrangement and packing of collagen fibrils within the stroma, as well as the unique composition of several substances such as KSPG (lumican, keratocan, mimecan, and decorin), stromal crystalline (transketolase, ALDH3A1, and ALDH1A1) and ECM proteins (collagen type I and V) [90]. When the cornea is damaged or diseased, the CSKs can die. They are responsible for producing proteoglycans and maintaining the collagen fibrils. Their death can lead to a reduction in the production of proteoglycans, the degradation of collagen fibers, and an increase in the glycation of collagen molecules. Nevertheless, some of the surviving CSKs can be activated and turned into repair-type stromal fibroblasts near the damaged area, contributing to the healing process of the cornea. The combined action of serum and cytokines such as PDGF and TGF β can cause certain fibroblasts to transform into highly contractile myofibroblasts. This mechanism can lead to the development of corneal haze, opacity, and scar formation. Such phenomena may disrupt the transmission of light through the cornea, which can result in visual impairment and ultimately lead to blindness.

Research has revealed that MSCs obtained from bone marrow and the lining of the umbilical cord can transform into keratocyte-like cells and potentially restore clarity to the corneal stroma [56,61,63]. A study involving the administration of human umbilical cord lining MSCs directly into the corneal stroma found improvement in the abnormal collagen structure, restoration of the corneal thickness, and enhancement of corneal transparency. In addition, the injected cells downregulated inflammatory cytokines, leading to a low risk of rejection [61]. In the same way, when human MSCs obtained from bone marrow, adipose tissue, and limbal stroma were cultured under conditions that promoted keratocyte differentiation and were supplemented with TGF β 3, basic fibroblast growth factor (bFGF), and ascorbic acid, genes associated with corneal stromal keratocytes were upregulated at the RNA and protein levels [56,73,91]. Similarly, in a study of mechanically induced corneal stromal defects, transplantation of rabbit adipose MSCs cultivated on a bio-scaffold made of polylactic-co-glycolic acid could repair the defects via the induction of ALDH1A1 and keratocan expression without triggering corneal neovascularization [74].

Another study showed that corneal stroma stem cells (CSSCs) derived from the limbal stroma share many characteristics with MSCs. These cells were found to express Pax6 and MSC markers (CD90 and CD73) [92,93]. CSSCs have the potential to repair and regenerate transparent stromal tissue as well as downregulate inflammation in the cornea and reduce scarring [93,94]. CSSCs can differentiate into CSKs when grown in a serum-free environment that has been enriched with bFGF and TGF β3. This suggests that CSSCs have the potential for stromal regeneration, with the deposition of an ECM similar to that of the native stroma [95,96]. Likewise, in the lumican-null mouse model, the injection of human CSSCs into the cornea repairs the defects in collagen fibrils and restores the stromal thickness, ultimately resulting in complete restoration of corneal transparency [93]. At the L.V. Prasad Eye Institute in India, medical treatment is being performed to address patients with unilateral superficial corneal scars resulting from bacterial/fungal keratitis. This procedure entails the transplantation of cultivated allogeneic limbal stromal cells [97].

Dental MSCs are gaining popularity in regenerative medicine because of their versatility, high adaptability, lack of ethical concerns, and ease of procurement [98]. The developmental pathways of CSKs are similar to those of periodontal ligament stem cells (PDLSCs) and dental pulp stem cells (DPSCs) [99]. Transplanting human DPSCs intrastromally into a mouse cornea resulted in the expression of collagen type I and keratocan, and they exhibited a phenotype resembling that of CSKs while preserving corneal transparency and maintaining the stromal volume [72]. PDLSCs exhibit multilineage potential and differentiate into adipocytes, chondrocytes, and osteoblasts. Additionally, they express markers present on MSCs and embryonic and neural stem cells [100,101,102]. A preliminary clinical study was conducted using stromal cell therapy, in which patients with advanced keratoconus were treated with autologous ADSCs. The cells were injected with 1 mL of saline into the pocket of the corneal stroma. New collagen was produced, and thus, the use of autologous ADSCs for cellular therapy for human corneal stroma is regarded as safe [103]. In addition, a clinical study involving 11 patients with advanced keratoconus demonstrated positive outcomes after autologous MSC transplantation, with or without decellularized donor corneal stromal lamina sheets. Within three months after the surgery, all patients had fully regained their corneal transparency [104]. However, it is necessary to have a larger group of participants and to follow them up for a longer duration to validate the effectiveness of this treatment.

In another study, the reconstruction of corneal stroma was observed with human-processed lipoaspirate derived (PLA). This outcome proved that human ADSCs retained their morphology up to 10 weeks after transplantation. Similarly, differentiation of these cells into keratocytes was seen in the rabbit cornea 12 weeks after transplantation with the synthesis of collagens type I and VI [105]. Likewise, transplantation of MSCs derived from dental pulp into rat’s eyes induced the production of an extracellular stromal matrix consisting of collagen type I and keratocon [72].

### 4.3. Corneal Endothelium Reconstruction with MSCs

The corneal endothelium is a thin, single-cell layer. It is the innermost layer of the cornea, which forms the boundary between the stromal and anterior chambers. According to one study, around 38% of cases requiring corneal transplantation are attributable to issues with the corneal endothelium [11]. Maintaining clarity is among the primary functions of the corneal endothelium [106]. The corneal endothelium transports fluid from the stromal layer to the anterior chamber, and its cells have a limited ability to undergo cell division [107]. In the event of any obstruction or dysfunction in the corneal endothelium, there is a risk of gradual accumulation of fluid in the stromal and epithelial layers, which can result in corneal edema and ultimately lead to impaired vision and even blindness [108]. One of the important indicators for corneal transplantation is corneal endothelial dysfunction. Recently, there has been an increase in the popularity of using Descemet’s stripping automated endothelial keratoplasty (DSAEK) and Descemet’s membrane endothelial keratoplasty (DMEK) for the treatment of endothelial dysfunctions [109]. Researchers are also exploring in-cell therapy as an alternative to corneal transplantation, which is often limited by a shortage of donors, and they are working to overcome the obstacles of growing endothelial cells in culture. They have discovered that Rho-associated protein kinase (ROCK) has the potential to repair and regenerate corneal endothelial cells, making corneal endothelial transplantation a viable option. In the rabbit endothelial dysfunction model, by using a ROCK inhibitor called Y-27,632, corneal transparency was restored by the transplantation of corneal endothelial cells. Even though this study is based on an animal model, there is a high likelihood that this method will yield positive results in clinical studies [110]. A recent investigation transplanted human corneal endothelial cells (HCECs) into the corneas of patients with bullous keratopathy after in vitro culture. The cells were mixed with ROCK inhibitor and transferred into the anterior chamber in a volume of 30 microliters containing 1 million cells. After 24 weeks, the corneas showed an improvement in transparency and an increase in corneal endothelial cell density. This approach was declared a less invasive treatment option for bullous keratopathy [111]. However, many obstacles remain to be overcome before applying this method to large-scale clinical treatment. Acquiring a sufficient number of cells and ensuring their proper proliferation in the laboratory can be difficult due to variations in cell fate and MET [112].

For corneal endothelial replacement, using MSCs and a conditioned medium is a promising approach [113]. One study found that exposing HCECs to the conditioned medium from BM-MSCs resulted in changes in the morphological and phenotypic characteristics of the HCECs [107]. Similarly, in another study, a damaged endothelium was restored by transplanting BM-MSCs, which were grown on a glutaraldehyde crosslinked gelatin scaffold, into the endothelial layers of rabbit corneas [114]. Likewise, in a rabbit model, transplantation of HUC-MSCs cultured on type I collagen sheets into the corneas resulted in a reduction in edema and an increment in corneal transparency [76]. Another study involved co-culturing skin-derived precursors (SKPs) and B4G12 cells in a serum-free medium, which after 4 days resulted in the formation of corneal endothelial-like cells with characteristics similar to human CECs. The markers of endothelium, NA+/K+ ATPase and zonula occludens-1 (ZO-1), were expressed well, as confirmed by immunofluorescence staining. In vivo, these cells were transplanted into rabbits and monkeys after Descemet’s membrane was mechanically removed, resulting in an increase in corneal transparency and a sharper appearance of the cornea after just 7 days [115].

Recently, a study demonstrated that hMSCs can differentiate into cells similar to corneal endothelial (CE) cells. The differentiated cells were found to express the markers typically found in corneal epithelial cells, such as ZO-1, Na/K-ATPase, COL-8, and paired-like homeodomain transcription factor 2 (PITX2). The differentiation process was achieved via the use of Descemet membrane biomimetic microphotography [116]. In this study, researchers implanted Warton jelly-derived stem cells in a Descemet membrane that had a collagen-like topography to create an endothelium-like layer. The researchers found an increase in the expression of genes specific to endothelial cells (COL-8, ZO-1, Na/K-ATPase, and PITX2). These results suggest that Wharton jelly-derived stem cells possess the capability to transform into cells with endothelial characteristics. Furthermore, transplantation of these cells in ex vivo rabbit cornea signified the formation of functional endothelium and transparent cornea [117].

## 5. Fate of MSCs in Corneal Inflammation and Angiogenesis

MSCs are recognized for their ability to regulate angiogenesis and reduce inflammation, making them a promising treatment option for various corneal diseases. Many studies have indicated that topical or sub-conjunctival administration of BM-MSCs can reduce inflammation and angiogenesis in murine models with chemical injuries [48,50,57]. Incorporating MSCs into corneal tissue has been shown to decrease the infiltration of inflammatory cells and macrophages expressing CD68. This leads to a reduction in pro-inflammatory cytokines, including interleukin-2 (IL-2), IL-1, monocyte chemoattractant protein (MCP-1), and matrix metalloproteinase 2 (MMP2), as well as pro-angiogenic factors. MSC treatment can also increase the expression of molecules with anti-inflammatory effects, such as IL-6, IL-10, TGF-β1, and TSG-6, along with anti-angiogenic mediators, such as thrombospondin-1 (TSP-1) and pentraxin-3 [54,57]. By modifying the pro-inflammatory environment, the corneal epithelium can be restored, leading to the healing of ocular surface injuries in the damaged cornea [51].

MSCs express pro and anti-angiogenic factors, depending on the tissue microenvironment. In order to inhibit angiogenesis, MSCs increase TSP-1 by disrupting the signaling of CD47 and vascular endothelial growth factor (VEGF) receptor 2, and inhibiting the VEGF-Akt-Enos pathway. The release of TSP-1, a potential inflammatory cytokine with pro-angiogenic activity, not only induces endothelial cell apoptosis but also decreases the expression of MMP2 [118]. Recent studies have demonstrated the modulatory actions of the CSSCs derived from the limbal stroma in corneal inflammation and scarring [119]. One study implemented CSCs in an acute corneal wound mouse model, and the inhibition of neutrophils and a decline in the expression of fibrotic markers like tenascin C, alpha-smooth muscle actin (α-SMA), and secreted protein acidic and rich in cysteine (SPARS), mediated through the TSG-6 pathway, was observed [94]. Likewise, in murine corneas, corneal mesenchymal stromal cells also perform an anti-angiogenic role via expression of PEDF and soluble fms-like tyrosine kinase-1 (sFLT) and suppression of macrophage infiltration [59]. Suture-induced corneal neovascularization was inhibited with the intravenous administration of CMSC, which decreased the expression of various genes involved in angiogenesis, such as VEGF-C, VEGF-D, TEK, and mannose receptor C type-1 and 2, within the stromal matrix [120].

## 6. Corneal Transplantation with MSCs

Numerous studies have explored the functions of MSCs in enhancing the survival of grafts [118,121,122]. The immune-modulating and anti-inflammatory characteristics of MSCs make them a promising choice for the transplantation of corneal allografts. Since the cornea is considered an immune-privileged site, the survival rate of corneal transplants is higher than other types of solid organ transplant. Regardless of this, corneal graft immune rejection can sometimes lead to corneal allograft failure [123]. The maturation of B-cells can be prevented by MSCs, and they can also inhibit the release of cytokines by T-cells. This modulation helps to sustain tolerance of the allograft and promotes its survival by influencing the generation of regulatory T-cells [124,125]. MSCs have a cell surface glycocalyx that contains a high concentration of anti-inflammatory molecules such as TSG-6, versican, and pentraxin-3. These molecules are involved in regulating the inflammatory response of the host [125]. The initial immunomodulatory impacts of recipient-derived MSCs from a pig-to-rat model were examined in the context of penetrating keratoplasty. When allogeneic rat MSCs were applied topically, they caused T-cells to differentiate into Th2 cells. However, even though the researchers induced an increase in Th2-type cytokines using MSCs, they did not observe a notable improvement in the survival of pig corneal tissue transplanted into rats [126].

Similarly, in a study conducted using a rat corneal transplant model, the researchers demonstrated the immunosuppressive capabilities of MSCs. Specifically, they injected MSCs from the donor into the recipient rats at varying intervals and with different doses of cyclosporine A (CsA). Prolonged corneal graft survival and no corneal allograft rejection were associated with a post-operative infusion of MSCs, whereas a pre-operative infusion was ineffective. Furthermore, MSCs in allogeneic keratoplasty inhibited allogeneic T cell responses in both in vitro and in vivo rat models, suggesting MSCs prevent allograft immune rejection and increase the survival rate of allografts by upregulating the number of Tregs [127]. Similarly, in a study performed on a mouse model of corneal allotransplantation, the peri-transplant intravenous infusion of human MSCs led to the suppression of inflammation and decreased the activation of antigen-presenting cells in the cornea, thereby increasing the survival rate of allografts and decreasing the risk of immune rejection [122]. Despite the potential of MSCs as an alternative method to treat and prevent immune rejection after corneal transplantation, their effectiveness in addressing corneal allograft rejection in various animal models and clinical scenarios is still unclear.

## 7. Challenges of MSC Therapy

Despite tremendous applications in regenerative medicine, MSCs have some significant risks, and solving this issue is a major challenge [128]. The major problem associated with MSCs is cellular heterogeneity. MSCs sourced from different origins can result in inconsistent outcomes. Furthermore, an additional critical aspect of cell therapy is the various techniques used to obtain, isolate, and cultivate the cells, which can result in differing outcomes [129]. One of the considerable problems associated with insufficient reproducibility of experimental findings is the variation in MSC protocols [128]. Full documentation of the complete process of MSC isolation, sorting, ex vivo expansion, purification, phenotyping, and conducting follow-up examinations should be carried out to ensure reproducible clinical efficacy and outcomes [130]. Prior to the clinical settings, it is necessary to establish standardized protocols for the isolation and ex vivo preparation of MSCs. However, parameters such as age, genetic traits, and the medical history of donors can impede corneal cell therapy. For example, the limited number of MSCs acquired from such cases makes it difficult to utilize autologous transplantation in elderly patients [131]. Furthermore, some studies have demonstrated that treatment with MSCs may not be sufficiently safe, as they are not immune-privileged [132]. However, the cornea itself is immune-privileged, and this is less important when treating corneal conditions. Likewise, senescence and decreases in differential capacity can be observed in MSCs after the sixth passage; this may lead to alterations in gene expression profiles and cell morphology, which can have adverse effects during cell therapy [133,134]. Another critical concern with MSCs is their tendency to transform into different lineages during ex vivo expansion. Thus, controlling and documenting the entire MSC expansion procedure should be made mandatory to ensure reproducibility in the preparation of these cells [135]. Furthermore, ethical challenges should be addressed before undertaking clinical trials. These challenges involve minimizing harm, appropriately selecting and recruiting subjects, ensuring informed decision making through the consent process, and preventing therapeutic misconception [136]. Table 2 illustrates the advantages and disadvantages of MSCs and MSC-derived exosomes.

## 8. Paracrine Effect of MSCs

Within the realm of regenerative medicine, MSCs can have a therapeutic impact by producing soluble factors that aid in the regulation of tissue healing, inflammation, angiogenesis, and immune responses [82,140]. Most MSCs are naturally found in the filtering organs like the lungs, liver, and spleen. In one study, the injection of MSCs beneath the conjunctiva in corneas injured by alkali exposure facilitated the healing of corneal wounds, even though the MSCs remained localized in the subconjunctival space [51]. In the same way, when MSCs or conditioned media from MSCs were topically administered in a mouse model of corneal epithelial injury, they alleviated corneal inflammation, reduced the formation of new blood vessels, and facilitated wound healing [141]. The fact that the majority of the MSCs were present in the corneal stroma rather than the epithelium implies that the therapeutic impact of MSCs is accomplished via a paracrine mechanism rather than direct cell substitution. This mechanism involves the release of soluble factors from MSCs, which can be delivered through extracellular vesicles or exosomes [140,142,143].

## 9. Mesenchymal Stem Cell-Derived Exosomes

### 9.1. Exosome Biogenesis

Exosomes, which are a subset of extracellular vehicles (EVs), are membrane-bound and generated within the endosomal compartment of almost all eukaryotic cells [144]. Exosomes are present in a wide range of biological tissues and fluids, and they are found in almost all cells, tissues, and bodily fluids, including urine, blood, plasma, cerebrospinal fluid, sweat, breast milk, semen, gastrointestinal secretions, saliva, and amniotic fluid [145]. Unlike microvesicles, which are formed during apoptosis, exosomes are produced directly from the plasma membrane and apoptotic bodies, and they originate from endosomes [146]. Once exosomes are secreted into the extracellular space, they bind to multivesicular bodies (MVBs), which have intraluminal vesicles (ILVs), then merge with the plasma membrane. The formation and packaging of exosomes are managed by endosomal sorting complexes required for protein (ESCRT), which are attracted to MVBs. Furthermore, many other associated proteins, such as Tsg101, Alix, and VPS4, are also actively involved in this process [146]. The amount and composition of exosomes released from the same parental MSCs differ due to the influence of external factors that dictate the secretion of exosomes [33]. Exosomes are internalized by recipient cells in the local microenvironment or transported to distant regions via the circulatory system. The uptake of exosomes by target cells can occur through three main mechanisms: (1) interactions between ligands and receptors, (2) endocytosis by the recipient cells, and (3) direct fusion with the cell membrane [147] (Figure 3). Once the exosomes have merged with the recipient cells, their contents are released into the cytoplasmic space.

### 9.2. MSC-Derived Exosome Components

The molecular composition of exosomes is subject to considerable variability and is influenced by several factors such as the cell type that produces them, changes made to them, and the pathological environment in which they are formed. The biological roles that exosomes fulfill depend on the types of nucleic acids, proteins, and lipids that they contain. The composition of the exosomes has become well known in recent decades due to massive progress in the fields of biotechnology, including proteomics, lipidomics, transcriptomics, and bioinformatics, which provide a theoretical basis for the use of exosome therapy in treating various diseases [33]. Phosphatidic acid, cholesterol, sphingomyelin, arachidonic acid, prostaglandins, and leukotrienes are exosomal lipids. They are involved in the formation of exosomes and the maintenance of their biological stability [145]. Exosomes contain various multifaceted proteins involved in their production, such as synthenin, ALIX, TSG101, and ESCRT complex. In addition, exosomes carry membrane transporter and fusion proteins, including annexins, heat shock proteins, and Rab GTPase [149]. Again, exosomes are enriched with various nucleic acids, such as genomic DNA, cDNA, mitochondrial DNA (mtDNA), long coding RNAs (IncRNAs), circular RNAs (CirRNAs), microRNAs (miRNAs), and mRNAs [150]. They have an indispensable role in modulating biological processes and the epigenetic remodeling of cells.

MSCs have been demonstrated to produce a greater quantity of exosomes compared to other types of primary cells [151]. Proteomic analysis of MSC exosomes has revealed 1927 distinct proteins that possess various functions required for their characteristics and formation [152]. The exosomes derived from MSCs display commonly occurring exosome surface proteins, such as tetraspanin (CD81, CD63, and CD9), ALIX, and Tsg101. Furthermore, they also contain heat shock proteins (HSP90, HSP70, and HSP60). Similarly, MSC membrane proteins for several adhesion molecules (CD73, CD44, and CD29) are also expressed by exosomes [144]. Likewise, diverse types of nucleic acids including lncRNAs, miRNAs, and mRNAs are found in MSC-derived exosomes. The miRNAs are given a greater interest than others, as it is believed that they are a form of non-coding RNA; they are approximately 22 nucleotides in length and modulate post-transcriptional gene expression [153]. These miRNAs are crucial molecules in MSC exosomes since they are involved in several biological activities, such as cell differentiation, angiogenesis, apoptosis, and inflammatory pathways. Researchers have found that a single miRNA can regulate several messenger RNAs, and conversely, a single messenger RNA can be influenced by several miRNAs [154]. Thus, these complex networks demonstrate the potential of MSC exosomes to alter numerous functional, physiological, and pathological effects.

### 9.3. Isolation and Storage of Exosomes

Several methods are commonly used for isolating exosomes, such as ultracentrifugation, size exclusion chromatography (SEC), polymer precipitation, immune affinity capture, microfluidics, and ultrafiltration (UF). These techniques are utilized to separate exosomes from other components in biological samples and obtain a pure exosome population for downstream analysis. Among these, for separating exosomes, ultracentrifugation is the most commonly used method and is considered the gold standard approach [155]. Ultracentrifugation separates proteins, vesicles, cell debris, and cells into uniform suspensions based on their differential sedimentation rate. SEC uses the size of small-molecule–protein complexes as the basis for separation. UF separates exosomes according to their size [156]. Immune affinity chromatography separates exosomes based on their interactions with high-specific-affinity antibodies and antigens [157]. Polymer precipitation alters the solubility or dispersion of exosomes in body fluids or cell cultures to precipitate them from samples, usually through the use of polyethylene glycol or agglutinin [158].

To date, there is no standard universally accepted guideline for the proper storage of exosomes. The integrity of exosome lipid membranes and their therapeutic efficacy can be influenced by the buffer composition, storage temperature, and the number of freeze–thaw cycles. One study showed no alterations in the size of MSC exosomes or overall exosomal membrane integrity after −20 °C freeze–thaw cycles in PBS. However, the size of the vesicles was significantly reduced after being stored at 37 °C for two days and at 4 °C for three days [159]. Likewise, one study found that exosomes could be stored at −20 °C for six months without altering their biochemical activity [97]. Another study found that when neutrophil-derived exosomes were frozen at −20 °C, the vesicle size decreased but it did not at −80 °C [160]. Another study found that adding protease inhibitors to urinary exosomes prior to freezing at −20 °C did not halt the decline in exosomal biochemical activity. However, complete recovery of activity was observed after seven months of storage when the exosomes were frozen at −80 °C [161]. A recent study showed that the stability of the exosomal membrane and biochemical function was further improved by adding 25 mM of trehalose [162]. The findings from these studies have provided evidence that exosomes can maintain their functional stability even when stored for extended periods at relatively mild temperatures.

## 10. Therapeutic Promise of MSC-Derived Exosomes for Ocular Tissue

Several studies have provided evidence that MSC-derived exosomes have a significant influence on eye tissues. In a rat model of experimental autoimmune uveitis (EAU), injecting exosomes derived from HUC-MSCs around the eye reduced the movement and accumulation of immune cells, such as leukocytes, macrophages, and natural killer cells. This was accomplished by inhibiting the MCP1/C-C motif chemokine ligand 2 (CCL21) and MYD88-dependent pathways. In addition, these exosomes restored retinal function and stimulated the expression of Gr-1, CD4, CD68, CD161, and IL17 [163]. Likewise, in a study of laser-induced retinal injury, injecting exosomes derived from MSCs cultured from either umbilical cord or adipose tissue into the eye improved visual function and modified the pro-inflammatory environment. This was achieved by hindering the production of pro-inflammatory cytokines MCP1, intercellular adhesion molecule-1 (ICAM-1), and TNF- α [164]. Similarly, in a study conducted on diabetic rats with hyperglycemia-induced retinal inflammation, injecting HUC-MSC-derived exosomes directly into the eye (intravitreal injection) improved their visual condition. This was achieved by suppressing the high mobility group Box 1 (HMGB1) signaling pathway, aided by miR-126 [165]. Furthermore, in a study involving rats with retinal damage induced by blue light, injecting umbilical cord MSC-derived exosomes directly into the eye (intravitreal injection) resulted in a dose-dependent reduction in choroidal neovascularization. This suppression was achieved by reducing the production of VEGFA and inhibiting the NFkB pathway through the transfer of miR-16 [166,167]. In addition, in an experiment using a rat model of optic nerve crush, injecting exosomes derived from BM-MSCs directly into the eye (intravitreal injection) restored the growth of retinal ganglion cells. This restoration occurred through the activation of argonaute-2 signaling [168]. Other studies showed that intravenous administration of MSCs can restore retinal function in models of EAU and laser-induced retinal injury [169,170]. A clinical trial was conducted recently to explore the impact of intravitreal injection of HUC-MSCs-derived exosomes on patients with refractory macular holes. The trial involved five patients and showed the intervention resulted in both functional and anatomical recovery. However, one of the patients had a severe inflammatory response [171].

## 11. MSC-Derived Exosomes and the Cornea

Injuries to the cornea caused by chemical or thermal burns, traumatic injury, immune disorders, and hereditary conditions can lead to inflammation, neovascularization, scarring, and ulceration. Delayed and careless treatment may cause permanent blindness. However, MSC therapy is considered a suitable candidate to provide anti-inflammation, anti-angiogenesis, and immunomodulatory activities during treatment. Several studies have shown that the paracrine action of MSCs can greatly enhance the process of wound healing by regulating inflammation, angiogenesis, and tissue regeneration through various factors. Moreover, limited studies conducted in vitro and in vivo have shown the healing benefits of the bioactive molecules present in MSC exosomes on corneal injury models.

In an experiment where rabbit CSCs were cultured with ADSC exosomes, there was a robust increase in cell proliferation, reduced cell death, downregulation of MMPs, and a substantial deposition of new ECM molecules, specifically collagen [172]. Furthermore, in a mouse model of a superficial stromal wound, topical application of exosomes derived from CSSCs reduced corneal inflammation and scarring by inhibiting the entry of neutrophils through a TSG-6-dependent pathway and decreasing the activity of genes related to fibrosis such as actin alpha 2 (ACTA2), collagen type 3 alpha 1 (COL3A1), and acidic and rich in cysteine (SPARC) [94]. Likewise, exosomes derived from human corneal mesenchymal stromal cells have the ability to stimulate the recovery of damaged corneal epithelium in mice [173]. In addition, in a murine model of mucopolysaccharidosis, HUC-MSCs exosomes that carried β-glucuronidase were found to reduce the accumulation of glycosaminoglycans, thereby leading to a decrease in corneal haze [34]. Similarly, a study investigated the impact of MSC-derived factors on the function of keratocytes in vitro and found a significant improvement in the cells’ performance. Furthermore, the paracrine activity of MSCs enhanced the survival of keratocytes by inhibiting apoptosis [174]. A research study utilized exosomes from ADSCs loaded with miRNA 24-3p and incorporated into a modified hyaluronic acid hydrogel. The resulting miRNA 24-3p-rich exosomes (Exos-miRNA 24-3p) were observed to enhance corneal epithelial defect healing, cell migration, and maturation while inhibiting fibrosis and reducing the levels of inflammatory cytokines (CD163, and MMP9) in both in vitro and in vivo rabbit models of corneal epithelial defects caused by alkali burns [175]. Additionally, in a study with iPSC-MSCs exosomes combined with thermosensitive chitosan-based hydrogels (CHI hydrogel) in a rat cornea damage model using a trephine mold, the iPSC-MSCs exosomes promoted reconstruction of the damaged corneal epithelium and stromal layer. In vivo, this study documented the downregulation of mRNA expression of three major collagen types, collagen type I alpha (COL1A), collagen type V alpha 1 (COL5A1), and collagen type V alpha 2 (COL5A2). Additionally, this led to the inhibition of scar formation, and ECM decomposition was prevented by inhibiting the translocation-associated membrane protein 2 (TRAM2) [176].

In a rat model of corneal allograft rejection, transplanting MSC-exosomes resulted in a significant increase in graft survival time by inhibiting the infiltration of CD4+ and CD25+ T cells, reducing the levels of interferon-gamma (IFN-γ) and C-X-C motif chemokine 11 (CXCL11), and inhibiting the Th1 signaling pathway [177]. In addition, a study using exosomes derived from HUC-MSCs showed significantly enhanced in vitro proliferation and the migration of corneal epithelial cells in a corneal mechanical wound healing model in rats. This was attributed to downregulation of phosphatase and tensin homolog (PTEN) levels and activation of the PI3K/Akt signaling pathway through the transfer of miRNA-21, resulting in better corneal repair and regeneration [178]. In an in vitro study using MSC-EVs in HCECs, it was revealed that MSCs-EVs (5–20 × 103 MSC-EV/cell) were capable of regenerating damaged HCECs by decreasing the number of apoptotic cells [179]. Exosomal microRNA from ADSCs inhibited the expression of homeodomain-interacting protein kinase 2 (HIPK2) in rabbit corneal keratocytes, suppressing the transformation of rabbit corneal keratocytes into myofibroblasts [180]. When comparing MSCs-Exos and iPSCs exosomes, researchers found that iPSC exosomes showed a better effect in vitro by inducing proliferation, migration, and cell cycle progression, and inhibiting apoptosis [181]. This study found that BM-MSCs-derived exosomes could promote the expansion and movement of human corneal epithelial cells in a manner that depended on the dosage used. This effect was related to the activation of the p44/42 MAPK signaling pathway. In an animal experiment using a mouse alkali burn model, the injection of BM-MSCs-derived exosomes was observed to promote the healing of corneal damage by reducing inflammation and mitigating the overproduction of proteins associated with fibrosis (α-SMA) and vascularization (CD31) [182]. In a murine corneal damage model caused by alkali burns, scientists demonstrated promising results by topically administering BM-MSCs-derived extracellular vesicles (BMSC-EVs) embedded with methylcellulose. The BM-MSCs-EVs were found to regulate cell death, inflammation, and angiogenesis in the damaged tissue, leading to faster corneal regeneration [183]. Similarly, it was demonstrated that exosomes obtained from HUC-MSCs influenced autophagy in both human corneal epithelial cells and a mouse corneal injury model. In vitro, this treatment caused an increase in cell proliferation, migration, and upregulation of proliferating cell nuclear antigen (PCNA), cyclin-dependent kinase 2 (CDK2), cyclin A, and cyclin E expression. In addition, when combined with an autophagy activator, HUC-MSCs ameliorated corneal defects by decreasing apoptotic and inflammatory gene expression in vivo [184]. Likewise, it has been observed that MSC-derived exosomes can mitigate inflammation and cell death in the cornea following injury. This is supported by the decreased expression of proinflammatory molecules like TNF-α, IL-1β, IL-8, and NF-κB, as well as the pro-apoptotic protein Cas-8. Furthermore, these exosomes have shown the potential to reduce corneal angiogenesis by inhibiting the expression of pro-angiogenic factors (VEGF) and angiogenesis-associated genes (MMP-2 and MMP-9) [185]. Moreover, researchers have proven that exosome-mediated targeting of NF-κB c-Rel can effectively accelerate corneal wound healing. Using exosomes loaded with c-Rel-specific siRNA on the corneal surface as a topical treatment reduced the expression of inflammatory cytokines and accelerated corneal wound healing in mice, including in cases of diabetic corneal injury. The results of this study suggested that inhibiting c-Rel might be an effective treatment approach for managing a corneal injury [186].

In a study with a corneal epithelial defect model in diabetic mice, exosomes derived from BM-MSCs labeled with PKH-26 regenerated the corneal epithelium by inducing the proliferation, repair, and migration of corneal epithelial cells. Furthermore, infiltrating inflammatory cells were relatively diminished in the cornea treated with exosomes [187]. A study assessed the effectiveness of ADSCs compared to MSC-derived exosomes for treating corneal injuries induced by alkali in rats. Both ADSCs and exosomes had the potential to enhance corneal healing and prevent complications resulting from alkali burns. This is due to their ability to reduce inflammation and prevent angiogenesis. However, the researcher came to the conclusion that exosomes from MSCs could be an especially auspicious alternative since they were less risky than stem cells, and their diminutive size facilitated their penetration through biological barriers to reach specific organs [188]. Interestingly, in a recent study, when MSC exosomes were treated with a human cornea-on-a-chip (developed using microfluidic technology), they had a favorable outcome on corneal epithelial wound healing by reducing the release of MMP-2 protein and acting as an anti-angiogenetic factor [189]. Furthermore, ADSCs inhibited ECM formation, increased proliferation, and promoted the reverse migration of C-X-C chemokine receptor type 4 (CXCR4) neutrophils, which alleviated neovascularization. These findings suggest that ADSCs have an antiangiogenic effect during corneal wound healing and can be a potential treatment approach that reverses neutrophil migration [190]. In addition, it was observed that exosomes from CSSC downregulated the fibrotic gene (Acta2) expression, blocked neutrophil infiltration, reduced scar formation, and restored the corneal morphology by transferring miRNA to corneal cells [191]. Likewise, in a rat model of corneal scarring, topically administration of MSC exosomes alleviated corneal injuries by inhibiting angiogenesis, modulating the immune response, minimizing scar formation and enhancing the healing process [192]. These studies have demonstrated that MSC exosomes can enhance the treatment of corneal ocular surface diseases and congenital corneal metabolic disorders by promoting ECM formation and cell proliferation, migration, and survival (Figure 4 and Table 3).

## 12. Challenges and Future Perspectives

MSC therapy is considered a promising strategy in different pre-clinical studies for treating different types of corneal disorders, either by directly repairing the damaged tissue or communicating with other cells through soluble factors. The immune-privileged nature of corneas results in a lower risk of rejection and tumorigenesis.

Exosomes play a critical role as mediators in exchanging cellular information in various non-ocular models. The recipient cells take up the proteins and miRNAs present in the exosome cargoes, resulting in a reduction in inflammation, immune modulation, induction of angiogenesis, wound repair, and overall improvements in functional and biological recovery. The use of autologous serum eye drops has been commonly employed to manage various conditions like severe dry eyes, persistent epithelial defects, chemical injuries, and neurotrophic keratopathy [193]. MSC therapy can be a potential option for treating corneal injuries or diseases, and the topical application of MSC exosomes could be used as an additional therapy to address corneal inflammation and neovascularization. Exosome eye drops derived from MSCs may provide a safer alternative compared to cell-based therapies since they have a lower risk of immunological rejection, uncontrolled proliferation, tumor formation, toxicity, and higher bioavailability.

The commercially available autologous serum eye drops are non-specific, typically focus on lubrication and symptom relief, and hardly restore the immune homeostasis in the diseased cornea without actively promoting tissue repair. In a prospective clinical trial, after topical administration of MSC-derived exosomes to patients with a diseased cornea, there was a reduction in fluorescein scores, a longer rear film breakup time, and an increment in tear secretion [194]. In addition, to obtain the potential therapeutic and biological effects of MSCs, coupling MSC therapy with topical MSC exosomes would be a better option. Thus, with this increasing experimental evidence with attractive advantages over MSCs, MSC-derived exosomes would be the perfect candidate for cell-free therapy, having low immunogenicity and less risk of tumor formation. More than 850 unique proteins and 150 different miRNAs are delivered in the cargoes of MSC-derived exosomes to affect various pathways in the target cells [115].

Due to their cell-free nature, MSC-derived exosomes can potentially reduce issues associated with immunological rejection and the first-pass effect often observed with systemic MSC treatment. This makes MSC-derived exosomes a promising candidate for use as biological carriers of therapeutic agents. Despite the therapeutic potential demonstrated by MSC-derived exosomes in numerous studies, it is crucial to address various ethical concerns before their application in clinical settings. The fact that exosomes are derived from stem cells but are not actual cells presents a potential hurdle in defining legal classifications and obtaining approval from regulatory bodies in different countries for their clinical use [195].

Other challenges remain to be overcome. First, the absence of standardization of the methods used for the purification, separation, and profiling of MSC-derived exosomes on a global scale may result in contentious issues arising from varied laboratory investigations. The transportation, preservation, and storage of exosomes are vital aspects that should be seriously considered. One study showed that thawed freeze-drying could effectively purify exosomes and facilitate their transportation and long-term storage. However, it is necessary to conduct additional research to determine whether this process has any impact on the characteristics of the exosomes. Second, research has shown the exosomes derived from MSCs to be safe and effective for the treatment of corneal diseases. Regardless, the pharmacological characteristics like bioavailability, targeting, pharmacokinetics, and bio-distribution of exosomes should be studied for proper therapeutic applications of exosomes. It is essential to closely monitor and understand the precise mechanism of action of exosomes prior to their use in clinical applications. Third, another important consideration is determining the optimal route of exosome administration, whether local or systemic. It is also essential to establish the appropriate dosage and dosing intervals and to conduct potency assays to evaluate potential toxicity at different doses. Fourth, optimization of the culture methods and isolation techniques is vital. Finally, the healing efficacy of exosomes is mainly attributed to their inherent cargoes, such as RNA, DNA, lipids, and proteins, which should be investigated more thoroughly. Additional research is necessary to clarify the exact mechanism responsible for the regeneration process. Recent research has focused on factors that regulate miRNA sorting into exosomes. Still, further research is necessary to understand the significance of other components, such as IncRNA, cirRNA, lipids, and proteins, and their effects on the physiological functions of exosomes. Interestingly, recent research has proposed the use of exosome-mimicking nanovesicles (NVs) as a substitute for natural MSC-derived exosomes, as they can be produced in larger quantities. However, a suitable method for producing them in a highly efficient manner is still lacking [196]. Thus, a comprehensive examination should be carried out before applying MSC-derived exosomes in clinical trials to guarantee their safety.

## 13. Conclusions

In conclusion, both MSCs and MSC-derived exosomes have shown diverse functions in the treatment of corneal diseases. Utilizing exosomes as a cell-free treatment option may reduce the risk associated with cell therapy. Nevertheless, our comprehension of the mechanisms and operations of MSC-derived exosomes is insufficient and requires further study. Additional research is necessary to determine the optimal dosage, administration route, interval, and mechanism of action of these exosomes before they can be used in clinical trials. The low risk of tumor formation and their ability to regulate cell fate make MSC-derived exosomes a promising approach in regenerative medicine.

## Figures and Tables

**Figure 1 ijms-24-10917-f001:**
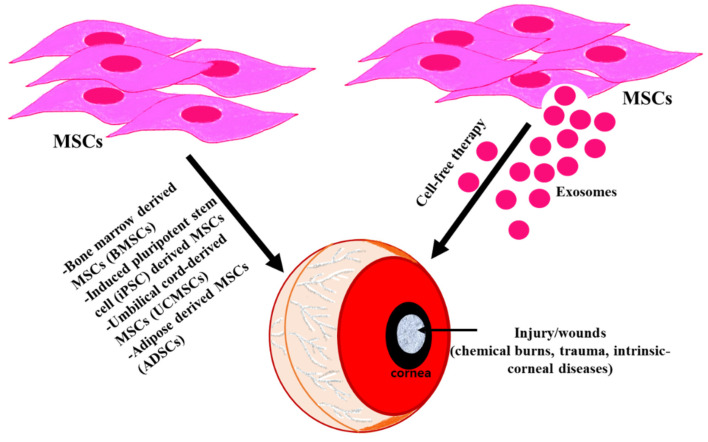
Cornea can be damaged from both external and internal factors (chemical burns, trauma, intrinsic–corneal diseases) leading to impaired vision. Different approaches have been employed to improve the regeneration of the injured cornea. Utilizing MSCs (cell-based therapy) and their secretions (cell-free therapy) can restore the normal corneal function in the diseased/injured cornea.

**Figure 2 ijms-24-10917-f002:**
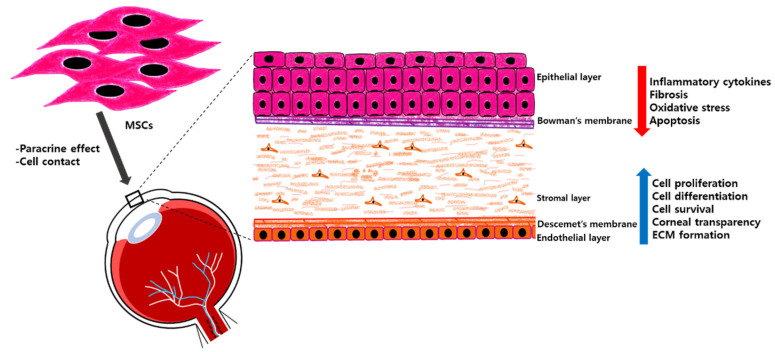
MSCs for treating corneal diseases. MSCs are well known to have a therapeutic effect through paracrine secretions, and by making direct cell contacts with cells in different corneal layers, such as epithelium, stoma, and endothelium. This can help to promote the survival, growth, and specialization of cells, while reducing cell death, inflammation, and fibrosis.

**Figure 3 ijms-24-10917-f003:**
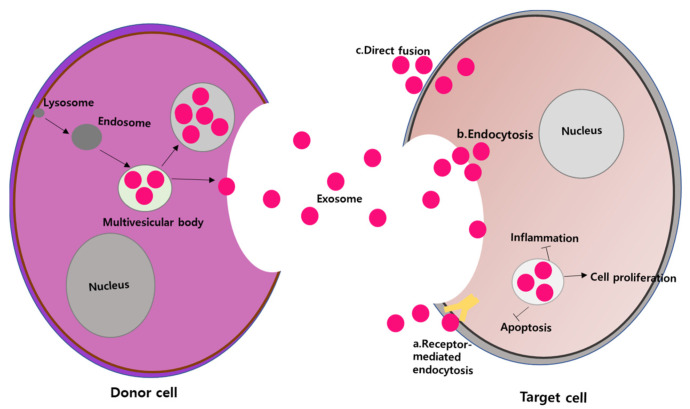
Exosomes are capable of transferring bioactive molecules to recipient cells through three mechanisms: (a) intercellular signaling via receptor–ligand interaction, (b) endocytosis by recipient cells, and (c) direct fusion with the recipient cell membrane, leading to the release of their cargo into the target cells. These modes of transfer are the result of the biogenesis of exosomes [97,148].

**Figure 4 ijms-24-10917-f004:**
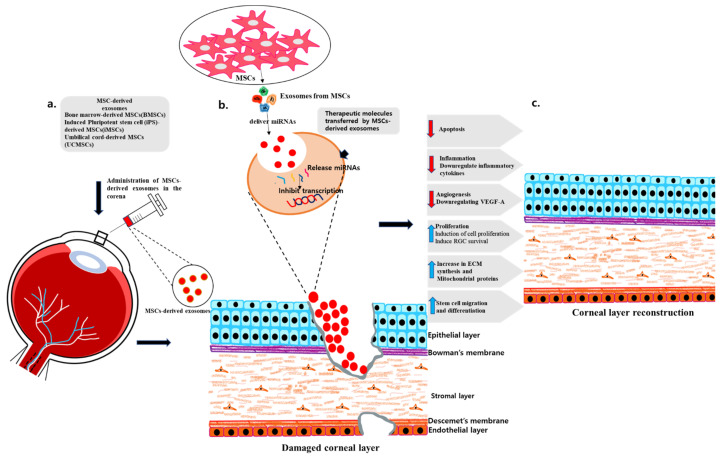
(**a**) A schematic representation of the experimental administration of MSC-derived exosomes in cornea. (**b**) In the cornea, therapeutic molecules are delivered that inhibit angiogenesis, cell proliferation, cell migration, and differentiation. (**c**) Regeneration of corneal layers is observed after treatment with MSC-derived exosomes.

**Table 1 ijms-24-10917-t001:** The therapeutic role of MSCs in different layers of cornea.

Corneal Tissues	MSC Source	Aim of Experiment	Human or Animal Model	Study Performed	In Vitro Study Results	In Vivo Study Results	References
	Human adipose	-To investigate the impacts of inhibiting glycogen synthase kinase-3 (GSK3) and transforming growth factor β (TGF β) signaling on the epithelial differentiation of ADSCs.	Rat	In vitro In vivo	-Downregulation of the mesenchymal genes and upregulation of epithelial genes (E-cadherin, cytokeratins, and occudin).	-Demonstration of the human E-cadherin CK3, and 12 on rat corneal surface in rat model of total limbal stem cell deficiency.	[66]
	Human adipose	-To study whether extraocular human ADSCs exhibit some characteristics of corneal epithelial-like cells cultured in vitro.	-	In vitro	-Induction of corneal epithelial-like cells from human adipose-tissue-derived MSCs including the expression of CK3 and CK12 when cultured in corneal epithelium-conditioned media.	-	[65]
	Rabbit bone marrow	-To study the potential of bone-marrow-derived MSCs (BM-MSCs) to differentiate into corneal epithelial cells both in vitro and in vivo.	Rabbit	In vitro In vivo	-Differentiation of rabbit MSCs vigorously into cells with similar morphological and molecular characteristics to corneal epithelial-like cells. -Induction of CK3 expression with co-culture with rabbit limbal stem cells.	-Implantation of the cells incorporated with fibrin gel regenerated the corneal epithelium in alkali-induced corneal deficiency rabbit model and expressed CK3.	[67]
	Rabbit bone marrow	-To inspect the suitability of bone marrow MSCs trans-differentiating into corneal epithelial cells in a rat model with a deficit of LSCs.	Rat	In vitro In vivo	-Observation of CK12 expression and epithelial cell characteristics with co-culture on rat corneal stromal cells.	-Differentiation into epithelial-like cells expressing CK12, improving corneal opacity, and reconstructing the corneal surface in rats with transplantation on amnion in alkali injury rat corneal deficiency model.	[53]
1. Corneal epithelium	Human bone marrow	-To explore the potential of human MSCs to differentiate into corneal epithelial cells and to assess their ability to regenerate damaged corneal tissue.	Rat	In vivo	-	- Restoration of the injured or damaged surface of the cornea in rats. -Inhibition of corneal inflammation (CD45, IL-2, MMP-2) and angiogenesis in the presence of amnion in rat alkali-injured epithelial defect model.	[57]
	Allogenic bone marrow	-To show that MSCs used in transplantation can be safe and effective and help in treating corneal pathology due to limbal stem cell deficiency (LSCD).	Human	In vivo	-	-Improvement of epithelial damage and demonstration of a more corneal epithelial-like phenotype in the central cornea with allogenic BM-MSCs.	[68]
	Rabbit bone marrow, adipose tissue	-To investigate whether MSCs or corneal limbal epithelial cells (LSCs) restore the corneal epithelium and optical properties in an alkali burn rabbit model.	Rabbit	In vivo	-	-Improvement in corneal optical properties, restoration of antioxidant protective mechanism and epithelial regeneration with MSCs.	[69]
	Conjunctiva-derived MSCs (CJMSCs)	-To identify the potential of hybrid polyurethane (PU) and silk nanofibrous scaffold with CJMSCs in treating corneal epithelium.	-	In vitro	-Improvement in the function of corneal epithelium with CJMSCs incorporated with silk fibers and PU fibers. -CJMSCs differentiate into corneal epithelial-like cells.	-	[70]
	Human bone marrow	-To demonstrate the aptitude of human MSCs derived from bone marrow to differentiate into functional cells with epithelial-like characteristics in vitro.	-	In vitro	-Differentiation of human MSCs derived from bone marrow into functional epithelial cells in the epithelial differentiation medium containing keratinocyte growth factor, epidermal growth factor, hepatocyte growth factor, and insulin-like growth factor.	-	[71]
	Human dental pulp	-To demonstrate the potential of adult dental pulp cells to differentiate into Keratocytes.	Mouse	In vitro In vivo	-Induction of Keratocan and Keratan sulfate proteoglycan (KSPG) in culture with keratocyte differentiation medium.	-Production of stromal components like human type-I collagen and Keratocan with intrastromal injection in mouse’s corneal stroma.	[72]
	Human adipose	-To determine whether keratocyte-specific phenotypic markers are expressed by ADSCs when cultured.	-	In vitro	-Induction of expression of stromal matrix components like KSPG, aldehyde-3-dehydrogenase-3A1 (ALDH3A1) in cell culture under reduced serum conditions supplemented with insulin and ascorbate.	-	[73]
2. Corneal stroma	Human bone marrow	-To investigate whether MSCs can differentiate into corneal keratocyte-like cells by using keratocyte-conditioned medium (KCM).	-	In vitro	-Expression of keratocyte markers such as aldehyde-1-dehydrogenase-1A1 (ALDH1A1), Lumican, and Kera in KCM. -Demonstrate MSCs could proliferate and differentiate into cells with similar characteristics to keratocytes when cultured in KCM.	*-*	[56]
	Rabbit adipose	-To explore if the combination of autologous rabbit adipose-derived stem cells and polylactic-co-glycolic acid (PLGA) scaffold could be used to repair corneal stromal defects in a rabbit.	Rabbit	In vivo	-	-Differentiation of MSCs into functional keratocytes, detection of their presence up to 24 weeks following transplantation. -Differentiation to Kera and ALDH3A1 expressing cells in mechanically induced rabbit stromal defect model via transplantation of cells on a PLGA scaffold.	[74]
	Human periodontal ligament (PDL)	-To show the possibility of using PDL cells as a potential source for regenerative corneal cell therapy to treat corneal disorders.	-	In vivo	-Organ culture shows the presence of CD34, ALDH3A1, Kera and Lumican, collagen type 8 alpha 2 (COL8A2), CHST6 genes and their expressions. -Reduction in fibrosis, neurogenesis, and vaculogenesis gene expression.	-	[75]
3. Corneal endothelium	Human umbilical cord MSCs (HUC-MSCs)	-To study the significance of HUC-MSCs in treating corneal endothelial disease in a rabbit model with bullous keratopathy.	Rabbit	In vivo In vitro	-Stimulation of the expression of NA, K-ATPase in a medium containing GSK3β inhibitor.	-Progress in corneal thickness and transparency with cell injection in rabbit bullous keratopathy model.	[76]

**Table 2 ijms-24-10917-t002:** Advantages and disadvantages of MSCs and MSC-derived exosomes in clinical applications [137,138,139].

Origin	Pros	Cons
MSCs	-Easy to isolate and obtain from accessible sources -High rate of proliferation, multilineal differentiation -Easily cultured in vitro -Low risk of immune-related problems -High stability in various pathological and physiological conditions	-Ethical issues -Risk of potentially transmitting genetic diseases and infections -Low number of cells -After transplantation, risk of teratoma formation is high
MSC-derived exosomes	-Capacity to cross natural barriers like blood–brain barrier -Perfect immune-compatibility and non-cytotoxic -Compared with cells, stable upon freezing and thawing -Ability of natural homing -Capacity for intracellular delivery of cargo by fusion of membranes	-No standard isolation protocol -No excellent mass production protocol -Less and immature research on exosome-based therapies -In vivo, after administration, fast clearance from the blood -Hard to isolate and purify the exosomes

**Table 3 ijms-24-10917-t003:** Preclinical studies of stem-cell-derived exosomes with their therapeutic potential for treating various corneal diseases.

Origin of Exosome	Aim of Experiment	Animal Model	Study Performed	In Vitro Study Conclusion	In Vivo Study Conclusion	References
ADSC exosomes	-To study how ADSCs exosomes may lead to phenotypic alterations in vitro in CSCs.	-	In vitro	-Inhibition of apoptosis, downregulation of MMPs, upregulation of ECM-related proteins (collagens and fibronectin) and significant proliferation of CSCs by ADSCs- exosomes.	-	[172]
Human corneal mesenchymal stromal cell-derived exosomes	-To investigate the impact of exosomes derived from human corneal MSCs on the healing of corneal epithelial wounds.	Mouse	In vitro In vivo	-Observation of corneal epithelial cell migration, proliferation, and adhesion. - Modulation of expression of genes related to cell signaling, inflammation, ECM remodeling.	-Improvement in corneal epithelial wound healing by augmenting cell proliferation.	[173]
BM-MSCs	-To examine how soluble factors derived from MSCs affect the functions of keratocytes (activation, migration, proliferation and synthesis of ECM).	Mouse	In vitro In vivo	-Enhancement of keratocye survival by inhibiting apoptosis, upregulation of ECM genes, increasing cell viability, and migration.	-Demonstration that various wound healing mediators like vascularly endothelial growth factor (VEGF), platelet-derived growth factor (PDGF), hepatocyte growth factor (HGF) accelerate corneal re-epithelization.	[174]
ADSCs exosomes	-To explore the potential of ocu-micro-RNA 24-3p to facilitate the migration and repair of rabbit corneal epithelial cells.	Rabbit	In vivo In vitro	-Increase in migration and proliferation of corneal epithelial cells.	-Promotion of rabbit corneal epithelial cell migration and repair by inhibiting fibrosis and keratitis, decreasing inflammatory reactions.	[175]
iPSC-MSCs exosomes	-To determine the effectiveness of exosomes obtained from iPSC-MSCs to repair damaged corneal epithelium and stromal layer, by decreasing the formation of scars and speeding up the healing process.	Rat	In vitro In vivo	-Downregulation of mRNA expression of COL1A, and COL5A2 in anterior lamellar stroma damage model in rats. -Reducing scar formation and regenerate corneal epithelium.	-Suppression of translocation-associated membrane protein 2 (TRAM2) by mi R-432-5p to prevent ECM decomposition.	[176]
MSC exosomes	-To study the effects of MSC-exosomes in corneal allograft rejection model.	Rat	In vivo	-	-Subconjunctival injection of 10 ug exosomes can effectively prolong the graft survival time. -Inhibit infiltration of CD4+ and CD25+T cells, downregulation of IFN-γ and CXCL11 levels in grafts.	[177]
HUC-MSCs-derived small extracellular vesicles (HUMSC-s EVs)	-To explore the mechanism through which HUMSCs-sEVs impact the healing process of corneal epithelial wounds.	Rat	In vitro In vivo	-Promotion of cell proliferation and migration via upregulating the P13k/Akt signaling pathway, achieved by restricting PTEN with transfer of miR-21.	-Corneal fluorescein staining and histological staining showed the healing of corneal wound in corneal mechanical wound rat model.	[178]
Stem cell-derived extracellular vesicles	-To explore the contribution of extracellular vesicles derived from stem cells in stem cell-induced regeneration, by reprogramming injured cells and triggering pro-degenerative pathways.	-	In vitro	-Decline in the quantity of apoptotic cells and faster wound repair in human corneal endothelial cells treatment with MSC-EVs.	-	[179]
ADSCs exosomes	-To examine the impact of exosomal miRNAs obtained from ADSCs on the differentiation process of rabbit corneal keratocytes.	-	In vitro	-Inhibition of HIPK2 expression suppresses the differentiation of corneal keratocytes into myofibroblasts. -Reduction in the expression of markers promoting pro-fibrosis and ECM components.	-	[180]
iPSCs exos and MSC exos	-To compare the efficacy of iPSCs-exos and MSC-exos in the treatment of corneal epithelial defects.	Rat	In vitro In vivo	-More promising result for iPSCs-exos than MSCs-exos. -Greater proliferation, migration, cell cycle progression, and inhibition of apoptosis in human corneal epithelial cells.	-Demonstration of stronger effects of iPSC-exos in healing corneal epithelial defect model than MSC-exos.	[181]
BM-MSCs exos	-To explore how BM-MSCs-exos promote corneal wound healing by activating the p44/42 MAPK signaling pathway.	Mouse	In vitro In vivo	-Enhancing the growth and migration of human corneal epithelial cells by BM-MSCs-exos.	-Downregulation of inflammation, fibrosis (α-SMA) fibrosis and vascularization (CD31) in corneal tissues of mice with alkali burn injury.	[182]
HUC-MSCs exos	-To explore the molecular mechanisms of HUC-MSCs-exos affect autophagy in vitro and corneal injury (CI) models in vivo.	Mouse	In vitro In vivo	-Combination of HUC- and MSCs-exos and an autophagy activator enhances cell proliferation, increases migration capacity, and boosts the expression of PCNA, Cyclin A, Cyclin E, and CDK2.	-Reduction in expression of apoptotic genes (Bax and Caspase 3), decrease in the inflammatory markers (TNF-α, IL-1β, IL-6 and CXCL-2) increment in BCL-2 in CI mice model.	[184]
HUC-MSCs	-To study whether the use of UMSC transplantation into corneal stroma has the potential to contribute in breakdown of glycosaminoglycans (GAGs), offering a viable approach for cell-based therapy for mucopolysaccharidoses (MPS)	Mouse	In vitro In vivo	-Release of neutral vesicles by HUC-MSCs which are taken by fibroblasts in a coculture assay	-Restoration of dendritic and hexagonal morphology of host keratocytes and endothelial cells, respectively, reduction in corneal haze. -Participation in breaking down extracellular GAGs and facilitating the host keratocytes to metabolize accumulated GAG products.	[52]
hMSCs exosomes	-To demonstrate the effectiveness of c-Rel-specific siRNA delivered through exosomes in accelerating corneal wound healing.	Mouse	In vivo	-	-Nano-polymers or exosomes containing c-Rel-specific siRNA on the corneal surface as a topical treatment help speed up the healing of corneal wounds in both regular and diabetic cases.	[186]
BM-MSCs-derived exosomes	-To examine how exosomes obtained from mouse BM-MSCs impact the regeneration of corneal epithelium in mice with diabetes.	Mouse	In vivo	-	-Restoration of corneal epithelial injury in diabetic mice by exosomes labelled with PKH-26 by downregulating the infiltration of cytokines and proliferation of corneal cells. -Detection of exosomes in the corneal stroma and nourishing it.	[187]
ADSCs and MSCs exosomes	-To assess and compare the effectiveness of ADSCs versus MSCs-exosome in treating corneal injuries induced by alkali in rats.	Rat	In vivo	-	-Improvement of corneal layers with decrease in inflammation and anti-angiogenic effects by MSCs-exosome-treated group in alkali burn injury model.	[188]
MSC exosomes	-To study the wound-healing and immunomodulatory effects of MSC exosomes in a rat corneal scarring model	Rat	In vitro In vivo	-Exerted immunomodulatory effect by modulating the expression and secretion of chemo-attractants.	-Improved the corneal epithelial wound healing by reducing corneal haze, supressing corneal neovascularization, downregulating inflamatory cytokines, and inhibiting angiogenesis in an excimer laser-induced rat corneal injury model.	[192]

## Data Availability

Not applicable.

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
