# Peer review of "Mesenchymal Stem Cells and Exosomes: A Novel Therapeutic Approach for Corneal Diseases"

_ijms, 2023, doi:10.3390/ijms241310917_

Round 1
Reviewer 1 Report
Very interesting review of a very important topic. Please consider discussing Djallian's work in this area. Also please discuss immune related work and potential pathways to abrogate rejection. For example microencapsulation. Please also discuss kill switches in setting of overgrowth. Please alao consider topical drops made out of msc exosomes as a replacement for serum tears. Discuss how this tech may be superior to other biologics. Ie prokera , AMG, etc
Bullous spelt wrong. Would try to break in more paragraphs for ease of read and cohesion of thought
Author Response
Reviewer 1 comments
#1 Very interesting review of a very important topic. Please consider discussing Djallian's work in this area. Also please discuss immune-related work and potential pathways to abrogate rejection. For example, microencapsulation. Please also discuss kill switches in the setting of overgrowth. Please also consider topical drops made out of MSC exosomes as a replacement for serum tears. Discuss how this tech may be superior to other biologics. Ie prokera , AMG, etc
Response #1
Thank you so much for the reviewer’s suggestions and great efforts toward improving our manuscript. We are really pleased to add this information stated by reviewers. We have discussed and referred the Djallian’s work in our updated manuscript. (Reference numbers: 2, 4, 14, 34, 79, 80, and 173). We have discussed the immune and pathways to abrogate rejection and about microencapsulation (Edit lines: 167-189, page numbers-5 and 6, section- Mesenchymal stem cells, Edit line:504-506, 513-516, page number-16, section-corneal transplantation with MSCs).
We have mentioned the kill switches in the setting of overgrowth (Edit lines: 190-207, page number- 6, section- Mesenchymal stem cells). We also have discussed the replacement and superiority of MSC-exosomes topical drops as a replacement for serum tears (Edit lines 841-855, page numbers-28 and 29, section challenges and future perspectives).
#2 Comments on the Quality of the English Language
Bullous spelled wrong. Would try to break it into more paragraphs for ease of reading and cohesion of thought.
Response #2
We are very grateful for the reviewer’s insightful comments on our paper. We have made corrections on the Bullous spelling (page number-10, table 1) and precisely checked in other sections also. We divided the single long paragraph into more paragraphs in each section for easy reading and understanding of the readers.
I gratitude for the reviewer’s advice.
With best regards,
Jae Yong Kim, MD, PhD
Professor
Department of Ophthalmology, Asan Medical Center,
University of Ulsan College of Medicine, Seoul 05505, Republic of Korea
Email: jykim2311@amc.seoul.kr
Reviewer 2 Report
Dear Authors,
This review articles very interesting. However it is very confusing and hard to follow and it requires a significant editing and restructuring to improve the its readability and the soundness and become publication worthy.
Thank you
The Reviewer
Dear Author,
The quality of your manuscript English language need to be improved. A proof reading by a native speaker is highly recommended.
Author Response
Reviewer 2 comments
#1 Comments and Suggestions for Authors
Dear Authors,
This review articles very interesting. However, it is very confusing and hard to follow and it requires a significant editing and restructuring to improve the its readability and the soundness and become publication worthy.
Response #1
Thank you so much for the reviewer’s comments and thoughtful words. We have made significant editing and reconstruction to improve its readability and soundness.
By respectfully following the reviewer’s suggestions, we split the long paragraphs into more paragraphs to improve the readability of our revised manuscript.
#2 Comments on the Quality of English Language
Dear Author,
The quality of your manuscript English language need to be improved. A proof reading by a native speaker is highly recommended.
Response #2
Thank you so much for the reviewer’s suggestions and great efforts toward improving our manuscript. We performed English proofreading and improved the quality of English in our updated version of the manuscript.
I gratitude for the reviewer’s advice.
With best regards,
Jae Yong Kim, MD, PhD
Professor
Department of Ophthalmology, Asan Medical Center,
University of Ulsan College of Medicine, Seoul 05505, Republic of Korea
Email: jykim2311@amc.seoul.kr
Reviewer 3 Report
Regarding to the manuscript entitled „Mesenchymal Stem Cells and Exosomes: A Novel Therapeutic Approach for Corneal Diseases” I share my comments below.
Pros:
Authors did an excellent work collecting plenty of information from variety of research and review articles and putting them together in presented manuscript.
Cons:
This manuscript is well organized and informative, however I find it slightly off the topic what makes it blurry in some parts. Furthermore, I feel like I read this somewhere, thus I do not find a novelty of this article.
Major issues:
The manuscript contains plenty of information on MSCs as well as EVs or exosomes. I do not see the reason why providing so much written knowledge into the new manuscript, while there are dozens of articles deeply explaining the biology of MSCs or EVs. Instead of providing so many details of common knowledge on both subjects It would be beneficial to focus on the main topic and follow the path from the aim to conclusion avoiding the off-topic details by citing excellent already published articles with same knowledge.
I find this article to possess enough material for two separate publications. Authors could consider shortening the contents or somehow modify and better adapt the topic of this work. Also, highlighting the part devoted to exosomes would raise the novelty of this article.
Minor issues:
line 75-77. Authors stated, “This is especially problematic given the increasing number of elderly individuals in our population, resulting in significant financial and logistical obstacles”. Authors could consider precising the population of which country is mentioned here or underline this as a global problem.
Line 94-97. I would recommend citing a reference to the statement “Today, ophthalmologists and visual scientists are increasingly interested in MSCs due to their competence to regenerate and differentiate, making them a potential alternative treatment option for corneal diseases. MSCs have been suggested to exhibit a therapeutic effect through their paracrine effect, which is mediated by exosomes (Figure 97 1).”. For example: PMID: 34620960
Figure 2. It is not clear how authors understand the “cell fusion” feature of MSC. It is also not supported by a source reference.
line 241. 50µ? I guess 50 µm?
line 258-260. Authors could consider citing more sources supporting this statement. For example: PMID: 31706096
Table 2. What are the disadvantageous ethical issues regarding using MSC in clinical applications?
Line 534. I would recommend moving this paragraph to the separate subsection describing the origin of EVs and them itself, because some of the authors highlight that EVs-based communication is not equal to paracrine activity.
Line 551. The spacebar mark is missing prior “Once the exsosomes...”
Figure 3. The caption information should be supported by references.
Line 599. I recommend stating ultracentrifugation than centrifugation.
Line 753. “stem-derived...” Stem cell-derived?
Author Response
Reviewer 3 comments
Comments and Suggestions for Authors
#1 Regarding to the manuscript entitled „Mesenchymal Stem Cells and Exosomes: A Novel Therapeutic Approach for Corneal Diseases” I share my comments below.
Pros:
Authors did an excellent work collecting plenty of information from variety of research and review articles and putting them together in presented manuscript.
Cons:
This manuscript is well organized and informative; however, I find it slightly off the topic what makes it blurry in some parts. Furthermore, I feel like I read this somewhere, thus I do not find a novelty of this article.
Response #1
Thank you so much for the reviewer’s insightful comments on our manuscript. We greatly appreciate your positive comments regarding its organization and informative nature. In terms of novelty, we appreciate your kind observation. Although there may be some similarities to previously published work, we believe our manuscript presents some extra perspective on this topic. We have referred to and discussed the recently published experimental data in our manuscript and tried to focus on the MSCs and MSCs-derived exosomes, their applications in corneal diseases, and future perspective and challenges.
Major issues:
#2 The manuscript contains plenty of information on MSCs as well as EVs or exosomes. I do not see the reason why providing so much written knowledge into the new manuscript, while there are dozens of articles deeply explaining the biology of MSCs or EVs. Instead of providing so many details of common knowledge on both subjects It would be beneficial to focus on the main topic and follow the path from the aim to conclusion avoiding the off-topic details by citing excellent already published articles with same knowledge.
I find this article to possess enough material for two separate publications. Authors could consider shortening the contents or somehow modify and better adapt the topic of this work. Also, highlighting the part devoted to exosomes would raise the novelty of this article.
Response #2
We appreciate the reviewer’s valuable feedback on our manuscript and thank you for sharing your perspective. We apologize if it appeared to be an excessive inclusion of common knowledge already covered in numerous existing articles. We acknowledge that the focus of our manuscript should primarily be on the main topic and the specific aim of our review.
In light of your feedback, we had carefully reviewed the manuscript and restructured it. In our manuscript, we mainly focused on MSCs and MSCs-derived exosomes because of their close relationship in terms of their therapeutic potential. By discussing both MSCs and MSC-derived exosomes together, we can explore their interconnected roles and highlight the importance of exosomes as mediators of MSC-based therapies. Further, we tried to give more information on their applications for corneal diseases. With their challenges and future perspectives in the field of regenerative medicine.
Respectfully following the reviewer’s kind suggestions, we split and arranged the different sub-headings into one main heading in some sections of our manuscript and omitted some of the extra unreliable sentences from the manuscript. We believe this approach makes the readers easy to understand and offers a distinct contribution to the scientific literature.
We sincerely appreciate your thoughtful recommendations and had adopted them in our manuscript, which will undoubtedly strengthen our manuscript.
Minor issues:
#3 line 75-77. Authors stated, “This is especially problematic given the increasing number of elderly individuals in our population, resulting in significant financial and logistical obstacles”. Authors could consider precising the population of which country is mentioned here or underline this as a global problem.
Response #3
Thank you so much for the reviewer’s comments. We have made changes in the sentences according to the reviewer’s suggestions. (Edit line:77, page number-2)
#4 Line 94-97. I would recommend citing a reference to the statement “Today, ophthalmologists and visual scientists are increasingly interested in MSCs due to their competence to regenerate and differentiate, making them a potential alternative treatment option for corneal diseases. MSCs have been suggested to exhibit a therapeutic effect through their paracrine effect, which is mediated by exosomes (Figure 97 1).”. For example PMID: 34620960
Response #4
We are grateful for the reviewer’s thoughtful comments. We have added the citation provided by reviewers and further added more citations. (Reference numbers: 16, 17, line number-99, page number-3)
#5 Figure 2. It is not clear how authors understand the “cell fusion” feature of MSC. It is also not supported by a source reference.
Response #5
Thank you so much for the reviewer’s suggestions. We have omitted the cell fusion from figure 2.
#6 line 241. 50µ? I guess 50 µm?
Response #6
We sincerely appreciate your thoughtful comments. We have corrected the sentence according to the reviewer’s suggestion (Edit line:285, Page number-11).
#7 line 258-260. Authors could consider citing more sources supporting this statement. For example: PMID: 31706096
Response #7
Thank you so much for the reviewer’s suggestions and great efforts toward improving our manuscript. We have cited the reference given by the reviewer and added more references (Reference number- 83).
#8 Table 2. What are the disadvantageous ethical issues regarding using MSC in clinical applications?
Response #8
We genuinely appreciate the reviewer’s great feedback. We have added the disadvantageous ethical issues in our original manuscript (Edit lines 545-548, page number- 17).
#9 Line 534. I would recommend moving this paragraph to the separate subsection describing the origin of EVs and them itself, because some of the authors highlight that EVs-based communication is not equal to paracrine activity.
Response #9
We are grateful for the reviewer’s comments. We have made corrections as stated by the reviewers in our main manuscript (Page number-18). We put this under the sub-heading Exosome Biogenesis.
#10 Line 551. The spacebar mark is missing prior “Once the exsosomes...”
Response #10
Thank you so much for the reviewer’s comments. We made the correction in the sentence. (Edit line 601, page number-19).
#11 Figure 3. The caption information should be supported by references
Response #11
Thank you for highlighting this aspect and offering valuable suggestions for the improvement of our manuscript. We have added the citation in the caption of the figure. (Edit line: 609, references numbers-97 and 148, page number-19).
#12 Line 599. I recommend stating ultracentrifugation than centrifugation.
Response #12
We are very grateful for the reviewer’s suggestions. We followed the reviewers’ suggestions and made corrections. (Edit lines:648, page number-20).
#13 Line 753. “stem-derived...” Stem cell-derived?
Response #13
Thank you so much for the reviewer’s comments. We made corrections to the sentence as stated by the reviewer. (Edit line: 818, page number-24).
I gratitude for the reviewer’s advice.
With best regards,
Jae Yong Kim, MD, PhD
Professor
Department of Ophthalmology, Asan Medical Center,
University of Ulsan College of Medicine, Seoul 05505, Republic of Korea
Email: jykim2311@amc.seoul.kr
Round 2
Reviewer 2 Report
Dear authors,
Thank you for revising the manuscript and making the necessary amendments/changes and working on the language which have resulted in a significant improve in the manuscript`s readability, comprehension and scientific soundness.
Best wishes
The Reviewer